# DEEP RANKING ENSEMBLES
# FOR HYPERPARAMETER OPTIMIZATION

**Abdus Salam Khazi**[*], **Sebastian Pineda Arango**[*], **Josif Grabocka**
University of Freiburg
Correspondence to Sebastian Pineda Arango: `pineda@cs.uni-freiburg.edu`

## ABSTRACT

Automatically optimizing the hyperparameters of Machine Learning algorithms is one of the primary open questions in AI. Existing work in Hyperparameter Optimization (HPO) trains surrogate models for approximating the response surface of hyperparameters as a regression task. In contrast, we hypothesize that the optimal strategy for training surrogates is to preserve the ranks of the performances of hyperparameter configurations as a Learning to Rank problem. As a result, we present a novel method that meta-learns neural network surrogates optimized for ranking the configurations' performances while modeling their uncertainty via ensembling. In a large-scale experimental protocol comprising 12 baselines, 16 HPO search spaces and 86 datasets/tasks, we demonstrate that our method achieves new state-of-the-art results in HPO.

## 1 INTRODUCTION

Hyperparameter Optimization (HPO) is a crucial ingredient in training state-of-the-art Machine Learning (ML) algorithms. The three popular families of HPO techniques are Bayesian Optimization (Hutter et al., 2019), Evolutionary Algorithms (Awad et al., 2021a), and Reinforcement Learning (Wu & Frazier, 2019; Jomaa et al., 2019). Among these paradigms, Bayesian Optimization (BO) stands out as the most popular approach to guide the HPO search. At its core, BO fits a parametric function (called a surrogate) to estimate the evaluated performances (e.g. validation error rates) of a set of hyperparameter configurations. The task of fitting the surrogate to the observed data points is treated as a probabilistic regression, where the common choice for the surrogate is Gaussian Processes (GP) (Snoek et al., 2012). Consequently, BO uses the probabilistic predictions of the configurations' performances for exploring the search space of hyperparameters. For an introduction to BO, we refer the interested reader to Hutter et al. (2019).

In this paper, we highlight that the current BO approach of training surrogates through a regression task is sub-optimal. We furthermore hypothesize that fitting a surrogate to evaluated configurations is instead a learning-to-rank (L2R) problem (Burges et al., 2005). The evaluation criterion for HPO is the performance of the top-ranked configuration. In contrast, the regression loss measures the surrogate's ability to estimate all observed performances and does not pay any special consideration to the top-performing configuration(s). We propose that BO surrogates must be learned to estimate the ranks of the configurations with a special emphasis on correctly predicting the ranks of the top-performing configurations.

Unfortunately, the current BO machinery cannot be naively extended for L2R, because Gaussian Processes (GP) are not directly applicable to ranking. In this paper, we propose a novel paradigm to train probabilistic surrogates for learning to rank in HPO with neural network ensembles [1]. Our networks are learned to minimize L2R listwise losses (Cao et al., 2007), and the ensemble's uncertainty estimation is modeled by training diverse networks via the *Deep Ensemble* paradigm (Lakshminarayanan et al., 2017). While there have been a few HPO-related works using flavors of basic ranking losses (Bardenet et al., 2013; Wistuba & Pedapati, 2020; Öztürk et al., 2022), ours is the first

---

[*]Equal contribution
[1]Our code is available in the following repository: `https://github.com/releaunifreiburg/DeepRankingEnsembles`

systematic treatment of HPO through a methodologically-principled L2R formulation. To achieve state-of-the-art HPO results, we follow the established practice of transfer-learning the ranking surrogates from evaluations on previous datasets (Wistuba & Grabocka, 2021). Furthermore, we boost the transfer quality by using dataset meta-features as an extra source of information (Jomaa et al., 2021a).

We conducted large-scale experiments using HPO-B (Pineda Arango et al., 2021), the largest public HPO benchmark and compared them against 12 state-of-the-art HPO baselines. We ultimately demonstrate that our method Deep Ranking Ensembles (DRE) sets the new state-of-the-art in HPO by a statistically-significant margin. This paper introduces three main technical contributions:

- We introduce a novel neural network BO surrogate (named Deep Ranking Ensembles) optimized with Learning-to-Rank (L2R) losses;

- We propose a new technique for meta-learning our ensemble surrogate from large-scale public meta-datasets;

- Deep Ranking Ensembles achieve the new state-of-the-art in HPO, demonstrated through a very large-scale experimental protocol.

## 2  RELATED WORK

**Hyperparameter Optimization (HPO)** is a problem that has been well elaborated on during the last decade. The mainstream HPO strategies are Reinforcement Learning (RL) (Wu & Frazier, 2019), evolutionary search (Awad et al., 2021b), and Bayesian optimization (BO) (Hutter et al., 2019). The latter comprises two main components: a surrogate function that approximates the response function given some observations, and an acquisition function that leverages the probabilistic output of the surrogate to explore the search space, ultimately deciding which point to observe next. Previous work covers various choices for the surrogate model family, including Gaussian Processes (Snoek et al., 2012), and Bayesian Neural Networks (Springenberg et al., 2016a). Other authors report the advantages of using ensembles as a surrogate, such as Random Forests Hutter et al. (2011), or ensembles of neural networks White et al. (2021). In contrast, we train BO surrogates using a learning-to-rank problem definition (Cao et al., 2007).

**Transfer HPO** refers to the problem definition of speeding up HPO by transferring knowledge from evaluations of hyperparameter configurations on other auxiliary datasets (Wistuba & Grabocka, 2021; Feurer et al., 2015; 2018). For example, the hyper-parameters of a Gaussian Process can be meta-learned on previous datasets and then transferred to new tasks (Wang et al., 2021). Similarly, a deep GP's kernel parameters can also be meta-learned across auxiliary tasks (Wistuba & Grabocka, 2021). Another method trains ensembles of GPs weighted proportionally to the similarity between the new task and the auxiliary ones (Wistuba et al., 2016). When performing transfer HPO, it is useful to embed additional information about the dataset. Some approaches use dataset meta-features to warm-initialize the HPO (Feurer et al., 2015; Wistuba et al., 2015), or to condition the surrogate during pre-training (Bardenet et al., 2013). Recent works propose an attention mechanism to train dataset-aware surrogates (Wei et al., 2019), or utilize deep sets to extract meta-features (Jomaa et al., 2021b). In complement to the prior work, we meta-learn ranking surrogates with meta-features.

**Learning to Rank (L2R)** is a problem definition that demands estimating the rank (a.k.a. relevance, or importance) of an instance in a set (Burges et al., 2005). The primary application domain for L2R is information retrieval (ranking websites in a search engine) (Ai et al., 2018), or e-commerce systems (ranking recommended products or advertisements) (Tang & Wang, 2018; Wu et al., 2018). However, L2R is applicable in diverse applications, from learning distance functions among images in computer vision (Cakir et al., 2019), up to ranking financial events (Feng et al., 2021). In this paper, we emphasize the link between HPO and L2R and train neural surrogates for BO with L2R.

**Learning to Rank for HPO** is a strategy for conducting HPO with an L2R optimization approach. There exist some literature on transfer-learning HPO methods that employ ranking objective within their transfer mechanisms. SCoT uses a surrogate-based ranking mechanism for transferring hyper-parameter configurations across datasets (Bardenet et al., 2013). On the other hand, Feurer et al. (2018) use a weighted ensemble of Gaussian Processes with one GP per auxiliary dataset, while the ensemble weights are learned with a pairwise ranking-based loss. Modeling the ranks of the learning

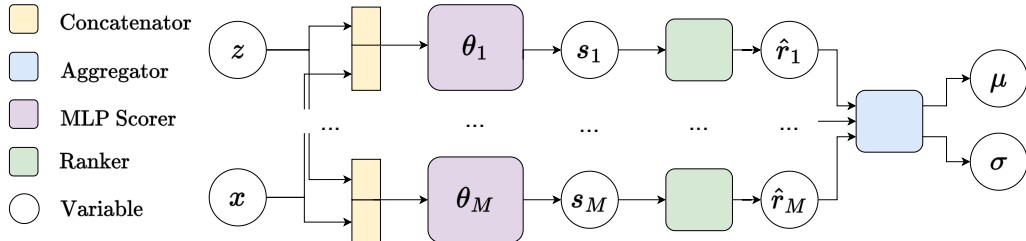

Figure 1: The neural architecture of our Deep Ranking Ensembles (DRE) with inputs $x$ (query points) and $z$ (meta-features).

curves also helps estimate the performance of configurations in a multi-fidelity transfer setup (Wistuba & Pedapati, 2020). Recent work has demonstrated that pair-wise ranking losses can be used for transfer-learning surrogates in a zero-shot HPO protocol (Öztürk et al., 2022). However, none of these approaches extensively study the core HPO problem with L2R, nor do they analyze which ranking loss types enable us to learn accurate BO surrogates.

## 3 DEEP RANKING ENSEMBLES (DRE)

### 3.1 PRELIMINARIES

**Hyperparameter Optimization** is defined as the problem of tuning the hyperparameters $x \in \mathcal{X}$ of a ML algorithm to minimize the validation error achieved on a dataset $D$ as $\arg\min_{x \in \mathcal{X}} \mathcal{L}^{\text{Val}}(x, D)$. The mainstream approach for tuning hyperparameters is Bayesian Optimization (BO), an introduction of which is offered by Hutter et al. (2019). BO relies on fitting a surrogate function for approximating the validation error on evaluated hyperparameter configurations. Consider having evaluated $N$ configurations on a dataset and their respective validation errors as $H = \{(x_i, y_i)\}_{i=1}^{N}$, where $y_i = \mathcal{L}^{\text{Val}}(x, D)$. We train a surrogate function $\hat{y}(x_i) = f(x_i; \theta)$, typically a Gaussian Process, to estimate the observed $y$ as $\arg\max_{\theta} \mathbb{E}_{(x_i, y_i) \sim p_H} \log p(y_i | x_i, H/\{(x_i, y_i)\}; \theta)$.

**Learning to Rank (L2R)** differs from a standard supervised regression because instead of directly estimating the target variable it learns to estimate the rank of the target values. In the context of HPO, we define the rank of a configuration as $r(x_i, \{y_1, \ldots, y_N\}) := \sum_{j=1}^{N} \mathbb{1}_{y_j \leq y_i}$. The core of a typical L2R method (Burges et al., 2005) includes training a parametric ranker $\hat{r}(x_i) := f(x_i; \theta)$ that correctly estimates the ranks of observed configurations' validation errors. Instead of naively estimating the ranks as a direct regression task, i.e. $\arg\max_{\theta} \mathbb{E}_{(x_i, r_i) \sim p_H} \log p(r_i | x_i; \theta)$, L2R techniques prioritize estimating the ranks of top-performing configurations more than bottom-performing ones (Cao et al., 2007). In general ranking losses can be defined on the basis of single objects (*point-wise approach*), pairs of objects (*pair-wise approach*) or the whole list of objects (*list-wise approach*) (Chen et al., 2009).

### 3.2 DEEP RANKING ENSEMBLE (DRE) SURROGATE

In this paper, we introduce a novel ranking model based on an ensemble of diverse neural networks optimized for L2R. We aim to learn neural networks that output the ranking score of a hyperparameter configuration $s : \mathcal{X} \to \mathbb{R}$. The ranks of the estimated scores should match the true ranks $\sum_{j=1}^{N} \mathbb{1}_{y_j \leq y_i} \approx \sum_{j=1}^{N} \mathbb{1}_{s(x_j; \theta) \geq s(x_i; \theta)}$, however, with a higher priority in approximating the ranks of the top-performing configurations using a weighted list-wise L2R loss (Cao et al., 2007). First of all, we define the indices of the ranked/ordered configurations as $\pi : \{1, \ldots, N\} \to \{1, \ldots, N\}$. Concretely, the $\ell$-th observed configuration is the $k$-th ranked configuration $\pi(\ell) = k$ if $k = \sum_{j=1}^{N} \mathbb{1}_{y_j \leq y_\ell}$. Ultimately, we train the scoring network using the following loss:

$$\arg\min_{\theta} \sum_{i=1}^{N} \mathcal{L}(x_i, y_i, y, \theta), \text{ where } \mathcal{L}(x_i, y_i, y, \theta) = w(\pi(i)) \frac{\exp^{s(x_{\pi(i)}; \theta)}}{\sum_{j=i}^{N} \exp^{s(x_{\pi(j)}; \theta)}} \qquad (1)$$

---

**Algorithm 1:** Meta-learning the Deep Ranking Ensembles

---

**Input** : Set of datasets $\mathcal{D}$, Number of iterations $J$, Number of ensemble scorers $M$
**Output:** DRE parameters $\theta_1, \ldots, \theta_M$, Meta-feature network parameters $\phi$

1   Initialize scorer networks with parameters $\theta_1, \ldots, \theta_M$ ;

2   Initialize the parameters $\phi$ of the meta-feature network $z$ from Jomaa et al. (2021a) ;

3   **for** $j = 1$ **to** $J$ **do**

4      Sample dataset index $i \in \{1, \ldots, D\}$, sample scorer network index $m \in \{1, \ldots, M\}$ ;

5      Sample a query set $H^{(s)} := \left\{ \left( x_1^{(s)}, y_1^{(s)} \right), \ldots, \left( x_{N^{(s)}}^{(s)}, y_{N^{(s)}}^{(s)} \right) \right\}$ from $\mathcal{D}_i$ ;

6      Sample a support set $H^{(z)}$ from $\mathcal{D}_i \setminus H^{(s)}$ ;

7      Compute meta-features $z(H^{(z)}; \phi)$ ;

8      Compute rank scores for the query set $s_i = s \left( x_i^{(s)}, z \left( H^{(z)}; \phi \right); \theta_m \right), i = 1, \ldots, N^{(s)}$ ;

9      Compute true ranks $\pi(1), \ldots, \pi\left(N^{(s)}\right)$ ;

10      Compute loss $\mathcal{L}(\pi, s; \theta_m, \phi)$ using Equation 1 ;

11      Update the meta-feature network $\phi \leftarrow \phi - \eta_\phi \frac{\partial \mathcal{L}(\pi, s; \theta_m, \phi)}{\partial \phi}$ ;

12      Update the ranker network $\theta_m \leftarrow \theta_m - \eta_{\theta_m} \frac{\partial \mathcal{L}(\pi, s; \theta_m, \phi)}{\partial \theta_m}$ ;

13   **end**

14   **return** $\theta_1, \ldots, \theta_M, \phi$ ;

---

The weighting functions $w : \{1, \ldots, N\} \to \mathbb{R}_+$ is defined as $w(\pi(i)) = \frac{1}{\log(\pi(i)+1)}$ and is used to assign a higher penalty to the top-performing hyper-parameter configurations, whose correct rank is more important in HPO (Chen et al., 2017). After having trained the scoring model of Equation 1 we estimate the rank of an unobserved configuration as $\hat{r}(\mathbf{x}; \theta) = \sum_{j=1}^{N} \mathbb{1}_{s(x_j; \theta) \geq s(\mathbf{x}; \theta)}$. Furthermore, Bayesian Optimization (BO) needs uncertainty estimates to be able to explore the search space (Hutter et al., 2019). As a result, we model uncertainty by training $M$ diverse neural scorers $s_1(x, \theta_1), \ldots, s_M(x, \theta_M)$ with stochastic gradient descent. The diversity of the ensemble scorers is ensured through the established mechanism of applying different per-scorer seeds for sampling mini-batches of hyperparameter configurations (Lakshminarayanan et al., 2017). Finally, the posterior mean and variance of the estimated ranks is computed trivially as $\mu(x) = \frac{1}{N} \sum_{i=1}^{N} \hat{r}(x; \theta_i)$ and $\sigma^2(x) = \frac{1}{N} \sum_{i=1}^{N} (\hat{r}(x; \theta_i) - \mu(x))^2$. The BO pseudo-code and the details for using our Deep Rankers in HPO are explained in Appendix A.

### 3.3 META-LEARNING THE DEEP RANKING ENSEMBLES

HPO is a very challenging problem due to the limited number of evaluated hyperparameter configurations. As a result, the current best practice in HPO relies on transfer-learning the knowledge of hyperparameters[2] from evaluations on previous datasets (Wistuba & Grabocka, 2021; Wistuba et al., 2016; Salinas et al., 2020). In this paper, we meta-learn our ranker from $K$ datasets assuming we have a set of observations $H^{(k)} := \left\{ \left( x_1^{(k)}, y_1^{(k)} \right), \ldots, \left( x_{N_k}^{(k)}, y_{N_k}^{(k)} \right) \right\}$; $k = 1, \ldots, K$ with $N_k$ evaluated hyperparameter configurations on the $k$-th dataset. We meta-learn our ensemble of $M$ Deep Rankers with the meta-learning objective in Equation 2, where we learn to estimate the ranks of all observations on all evaluations for all the previous datasets using the loss of Equation 1.

$$\operatorname*{arg\,min}_{\theta_1, \ldots, \theta_M} \sum_{k=1}^{K} \sum_{n=1}^{N_k} \sum_{m=1}^{M} \mathcal{L} \left( x_n^{(k)}, y_n^{(k)}, y^{(k)}; \theta_m \right) \qquad (2)$$

---

[2]Even in manually-designed ML systems, experts start their initial guess about hyper-parameters by transfer-learning the configurations that worked well on past projects (a.k.a. datasets).

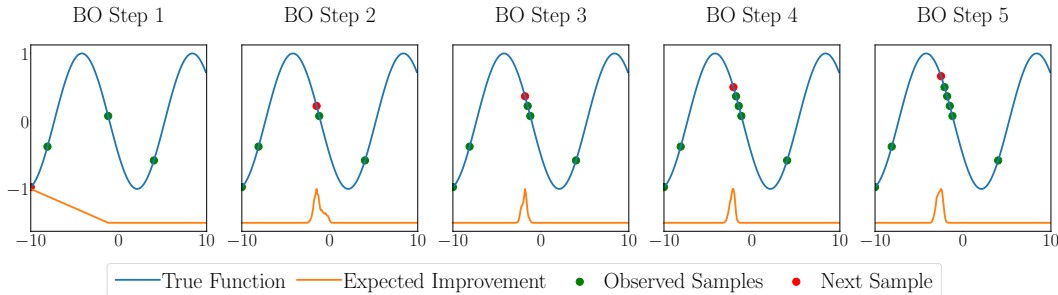

Figure 2: BO Steps Example with a Random Initialized DRE. EI is scaled and shifted for clarity.

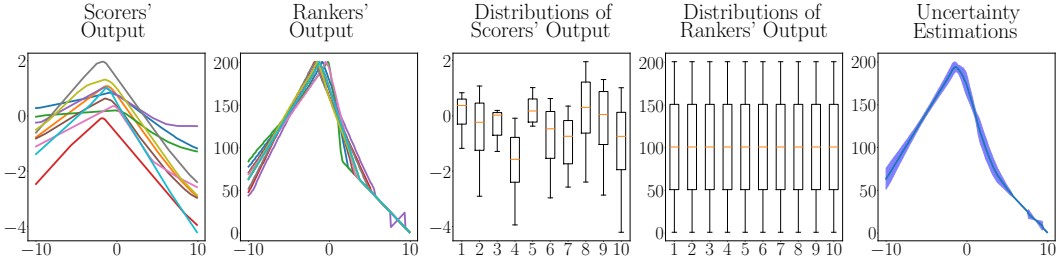

Figure 3: Understanding the outputs of DRE's modules.

Transfer-learning for HPO suffers from the negative-transfer phenomenon, where the distribution of the validation errors given hyperparameters changes across datasets. In such cases, using dataset meta-features helps condition the transfer only from evaluations on similar datasets (Rakotoarison et al., 2022; Jomaa et al., 2021a). We use the meta-features of Jomaa et al. (2021a) which are based on a deep set formulation (Zaheer et al., 2017) of the pairwise interactions between hyperparameters and their validation errors. The meta-feature network with parameter $\phi$ takes a history of evaluations $H = \{(x_i, y_i)\}_{i=1}^{N}$ as its input and outputs a $L$-dimensional representation of the history as $z(H, \phi)$ : $(\mathcal{X} \times \mathbb{R})^N \to \mathbb{R}^L$. Afterward, the scorer function becomes $s(x, z(H; \phi); \theta) : \mathcal{X} \times \mathbb{R}^L \to \mathbb{R}$. In other words, the dataset meta-features are additional features to the scorers. A graphical depiction of our architecture is shown in Figure 1.

We update all the scorer networks of the ensemble independently using the loss of Equation 1. The pseudo-code of Algorithm 1 draws an evaluation set (called a query set) which is used as the training batch for updating the parameters of the sampled scorer network. We also meta-learn the meta-feature network (Jomaa et al., 2021a), however, by using a different batch of evaluations (called a support set). We do not meta-learn both the scorer and the meta-feature networks using the same batch of evaluations in order to promote generalization.

## 4 EXPERIMENTS AND RESULTS

### 4.1 MOTIVATING EXAMPLE

We demonstrate our DRE with 10 base models on a simple sinusoid function $y = \sin(\frac{x+\pi}{2})$ for $x \in [-10, 10]$ sampled with a step size of 0.1 in equally spaced intervals. Further details on the architecture are explained in Section B. In Figure 2, we conduct BO with a variant of DRE without meta-learning, and we start the HPO with 3 initial random observations. We observe that a BO procedure with the Expected Improvement acquisition reaches an optimum after 8 observations.

Furthermore, we plot the scorers' and rankers' outputs of the second BO step in Figure 3. The analysis illustrates that the distributions of the scorers' outputs have different ranges because the loss function in Equation 1 models only the target rank, but not the scale of the target. However, the outputs of the rankers display similar distributions in the rank space, which is more adequate for

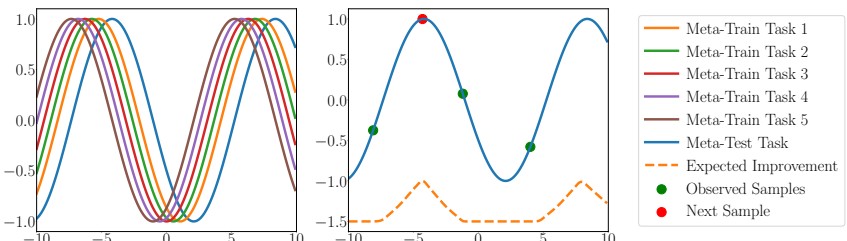

Figure 4: Meta-train and Meta-test Tasks (left) for optimizing the function. The meta-learned DRE finds the optimum in one step (right).

computing the ranks' uncertainties. Moreover, the rank distributions differ in certain regions of the search space, enabling BO to conduct exploration.

To showcase the power of transfer-learning, we meta-learn DRE on 5 auxiliary tasks, corresponding to different sinusoidal functions $y = \sin(\frac{x+\pi}{2} + \beta)$ with varying $\beta \in \{11, .., 15\}$, as illustrated in Figure 4 (left). Subsequently, we deploy a meta-learned DRE surrogate on a test task (blue line with $\beta = 8$) which was not part of the meta-training set. Figure 4 reveals that DRE directly discovers a global optimum within one BO step (4 total observations). The success is attributed to the fact that the surrogate has been meta-learned to recognize sinusoidal shapes given the 3 initial observations in green, as is clearly shown by the acquisition in Figure 4 (right).

## 4.2 DATASETS AND BASELINES

We base our experiments on HPO-B (Pineda Arango et al., 2021), the largest public benchmark for HPO. It contains 16 search spaces, each of which comprises a meta-train, meta-test, and meta-validation split. Every split is a set of datasets, and for every dataset, the benchmark contains the validation errors of evaluated hyperparameter configurations. The benchmark also includes the results of several HPO methods run in those datasets, including transfer and non-transfer algorithms [3]. Moreover, we generated new results for three additional state-of-the-art baselines (GCP, HEBO, and DKLM) that are not released by HPO-B. The benchmark provides 5 sets of 5 initial random seeds for every task in the meta-test split (86 in total). We use the meta-test datasets to compare the performance of the Deep Ranker Ensembles against the baselines. Specifically, our non-transfer HPO baselines are listed below:

- **Random Search (RS)** (Bergstra & Bengio, 2012) is a simple yet strong baseline that selects a random configuration at every step.
- **Gaussian Processes (GP)** (Snoek et al., 2012) model the response function by computing the posterior distribution of functions induced by the observed data.
- **DNGO** (Snoek et al., 2015) uses a neural network that models the uncertainty with a Bayesian linear regression on the last network layer.
- **BOHAMIANN** (Springenberg et al., 2016b) is also a Bayesian neural network that performs Bayesian inference via Hamiltonian Monte Carlo.
- **Deep-Kernel Gaussian Processes (DKGP)** (Wilson et al., 2016) learn a latent representation of the features that are fed to a GP kernel function.
- **HEBO** (Cowen-Rivers et al., 2020) is a state-of-the-art Bayesian optimization method. It combines input and output transformations and a multi-objective acquisition function. We use the implementation contained in the original repository.[4]

Transfer HPO methods use the evaluations of the tasks included in the meta-train split to meta-learn surrogates, that are subsequently applied for HPO on the meta-test tasks within the same search space. We consider the following baselines:

---

[3]Available in `https://github.com/releaunifreiburg/HPO-B`
[4]Available in `https://github.com/huawei-noah/HEBO`

- **TST** (Wistuba et al., 2016) constructs an ensemble of Gaussian Processes aggregated with a kernel-weighted average. Alternatively, **TAF** builds an ensemble of acquisition functions.

- **RGPE** (Feurer et al., 2018) trains a Gaussian Process per each meta-train task and then combines for a new task through a weighting scheme, which accounts for the ranking performance of every base GP model.

- **FSBO** (Wistuba & Grabocka, 2021) pre-trains a Deep Kernel Gaussian Process using meta-train tasks and then fine-tunes the parameters when observations for new tasks are available.

- **GCP** (Salinas et al., 2020) pre-trains a neural network to predict the residual performance on the auxiliary tasks and applies Gaussian Copulas to combine results for a new task.

- **DKLM** (Jomaa et al., 2021b) adds a Deep Set as task contextualization on top of FSBO. We use the same hyperparameters as suggested in the original paper.

## 4.3 DRE-Experimental Setup

The meta-feature extractor $z$ is based on the Deep Set architecture proposed by Jomaa et al. (2021a) with five hidden layers and 32 neurons per layer. The ensemble of scorers is composed of 10 MLPs with identical architectures: four layers and 32 neurons that we selected using the meta-validation split from HPO-B. We meta-learn DRE for 5000 epochs with Adam optimizer, learning rate 0.001 and batch size 100. Every element of the batch is a list of 100 elements. We select 20% of the samples in each list as input to the meta-feature extractor. During meta-test in every BO iteration, we update the pre-trained weights for 1000 epochs. For DRE-RI, we initialize randomly the scorers and train them for 1000 epochs using Adam Optimizer with a learning rate of 0.02. Every epoch, we use 20% of the observations to feed the meta-feature extractor.

## 4.4 Research Hypothesis and Experimental Results

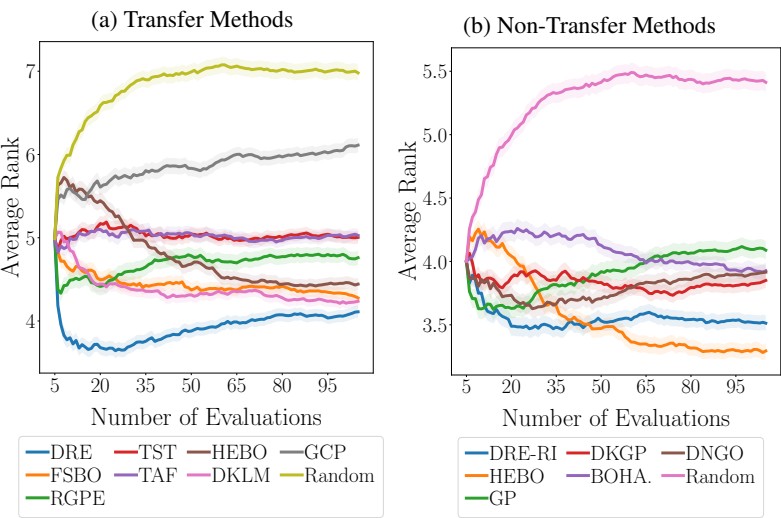

Figure 5: Results for Transfer and Non-transfer methods.

**Hypothesis 1.** *Deep Ranking Ensembles (DRE) achieve state-of-the-art results in transfer HPO.*

We compare against the transfer HPO baselines listed in Section 4.2 and report the average ranks across all the tasks in the meta-test split of all the HPO-B search spaces. Our protocol uses 5 initial configurations plus 100 BO iterations across 16 search spaces (the default HPO-B protocol). Our method uses meta-features (Jomaa et al., 2021a) and the scorer parameters are fine-tuned after each BO observation.

Figure 5 (left) shows that DRE clearly outperforms all baselines over 100 BO iterations based on the rank among the HPO methods averaged among 86 datasets and 5 runs. We compute the critical difference diagram (Demšar, 2006) for 25, 50, and 100 iterations, and show the statistical significance

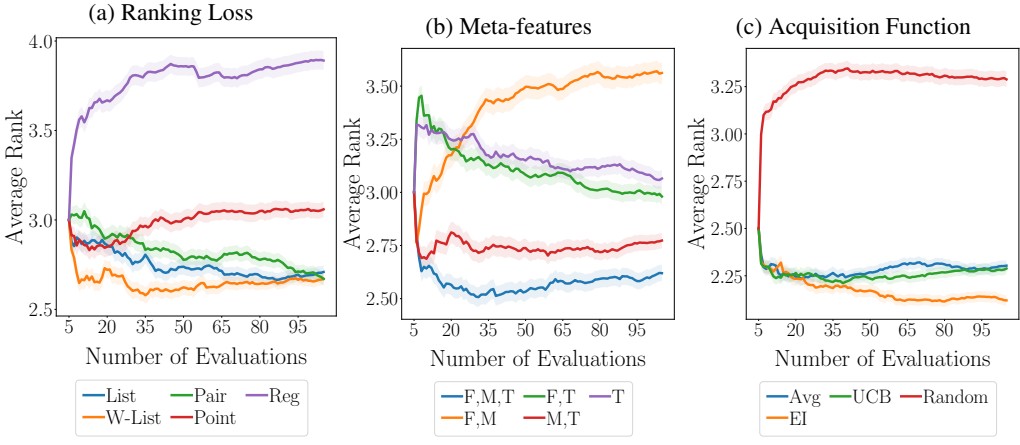

Figure 6: Results after testing our hypothesis 3-5.

of the results in Figure 9a (Appendix E). HEBO is not a transfer HPO method but is presented as a reference. These results demonstrate the advantage of training neural ensembles with L2R since our method outperforms other rivals which also meta-train neural networks (FSBO, DKLM), or ensembles of neural networks (TST, TAF, RGPE). DRE also attains competitive results in individual search space, as shown in Figure 13, at Appendix E.

**Hypothesis 2.** *The randomly-initialized DRE performs competitively in non-transfer HPO.*

We test the hypothesis by comparing the performance of DRE against the non-transfer baselines mentioned in Section 4.2. Similar to Experiment 1, we compute the average rank over 100 BO iterations, aggregating across all the meta-test tasks of all the search spaces in HPO-B.

The results of Figure 5 (right) show that a random initialized DRE (i.e. non meta-learned) is still a competitive surrogate for HPO. It exhibits good performance for up to 30 iterations compared to the other baselines and is second only to HEBO (notice our meta-learned DRE actually outperforms HEBO, Figure 5 (left)). This demonstrates the usefulness of deep ensembles with L2R as general-purpose HPO surrogates. Interestingly, DRE outperforms other surrogates using neural networks, such as BOHAMIANN, DNGO, and DKGP. We present the statistical significance of the results after 25, 50, and 100 BO iterations in Figure 9b.

**Hypothesis 3.** *A weighted list-wise ranking loss is the best L2R strategy for DRE.*

We test DRE (meta-learned) with three different L2R losses: point-wise, pair-wise, and list-wise (weighted and non-weighted) ranking losses. Additionally, we compare to a surrogate predicting the performance in the original scale using Mean Squared Error as loss, i.e. a regression. Moreover, we compare the performance to a DRE trained with a regression loss. We omit the meta-features from all variants to avoid confounding factors from the analysis and use Expected Improvement as the acquisition function.

The results in Figures 6a and 11a (Appendix E) show the advantage of the list-wise ranking losses over the other type of ranking losses. Moreover, the results highlight the advantage of list-weighted ranking losses, as it attained the best performance over the average rank among 100 BO iterations. Additionally, we observe that pairwise-losses also give a boost in performance compared to point-wise estimations. The message is: "Any L2R loss is better than the regression one".

**Hypothesis 4.** *Meta-features help the transfer HPO performance of DRE.*

We evaluate DRE with and without the meta-features extracted by the DeepSet module (Jomaa et al., 2021a), ablating the scenarios with and without meta-learning. Again we use all 16 search spaces from HPO-B for 100 BO iterations, starting with 5 random initial configurations. DRE uses the weighted list-wise loss, and Expected Improvement as the acquisition.

Figure 6b shows the performance obtained with meta-features (F) considering meta-learning (M) and fine-tuning (T). A missing capital letter in the label stands for an experiment without that aspect (e.g. no M means no meta-learning, etc). The results indicate that the meta-features help DRE achieve better performance, both with and without meta-learning. The results also highlight that fine-tuning (i.e. updating the scorer network's parameters on the target tasks after each BO step) the meta-learned surrogate is important for achieving the best HPO performance. Further evidence of the significance of these results is showcased in Figure 11b (Appendix E).

**Hypothesis 5.** *Expected Improvement is the best acquisition function for BO with DRE.*

We run experiments to address how DRE performs with different acquisition functions, which use DRE's estimated rank uncertainty to explore the search space with Bayesian Optimization. Concretely, we ablate the Upper Confidence Bound (UCB) and Expected Improvement (EI) acquisitions. Additionally, we added Average Rank (Avg) which simply recommends the configuration with the highest estimated average rank, without using the posterior variance of the rank. We also add Random Search as a reference baseline. Further details on how we apply acquisitions in the BO loop are discussed in Appendix A. In this experiment, we use meta-features and weighted list-wise ranking losses.

The results in Figures 6c and 10b (Appendix E) demonstrate that EI is the best choice for the acquisition function. As UCB and EI attained overall better performances than the simple average rank (no uncertainty), we conclude the uncertainties computed by DRE are effective in exploring the search space.

## 4.5 DISCUSSION ON DRE HYPERPARAMETERS

Given that DRE achieves state-of-the-art results across all the 16 search spaces (see Figure 12 in Appendix E) of HPO-B by using the same configuration (e.g. number of layers for the scorers, number of layers for meta-feature extractor), we assume our settings (hyper-hyperparameters) are applicable straightforwardly to new search spaces. Such a generalization of the hyper-hyperparameters is desirable for any HPO method and liberates practitioners and researchers from having to tune DRE hyper-hyperparameters. In Figure 7 we show an ablation study comparing the performance of DRE for different numbers of layers (2, 3, 4), and different numbers of neurons per layer (16, 32, 64) on all the tasks of the meta-validation split from HPO-B. Given the critical difference diagram in Figure 10a, we observe the performance does not change significantly when we vary any of these hyper-hyperparameters. However, we notice that the depth of the scorer is slightly more important to tune than the number of neurons per layer. We also notice that even an expressive ensemble of scorers (32x4) is able to generalize well on the meta-test split, as we have shown in our previous experiments.

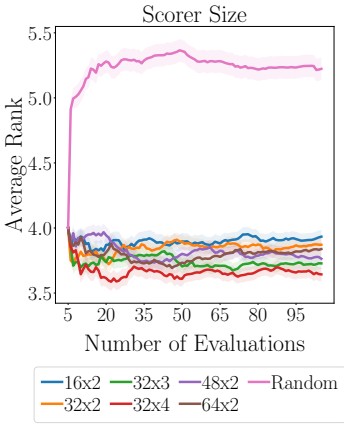

Figure 7: Average Rank on the meta-validation split from HPO-B.

## 5 CONCLUSION

The presented empirical results based on a very large-scale experimental protocol provide strong evidence of the state-of-the-art performance of deep ensembles optimized through learning to rank. We demonstrated that our method outperforms a large number of 11 baselines in both transfer and non-transfer HPO. In addition, we validated the design choices of our method through detailed ablations and analyses. Particularly, the results indicate the power of meta-learning surrogates from evaluations on other datasets. Overall, we believe that this paper will set a new trend in the HPO community for moving away from regression-learned surrogate functions in Bayesian Optimization. Finally, our surrogate DRE opens up an effective way to improve the HPO performance in different sub-problems, such as multi-fidelity HPO, multi-objective HPO, or neural architecture search.

ACKNOWLEDGEMENTS

This research was funded by the Deutsche Forschungsgemeinschaft (DFG, German Research Foundation) under grant number 417962828 and grant INST 39/963-1 FUGG (bwForCluster NEMO). In addition, Josif Grabocka acknowledges the support of the BrainLinks- BrainTools Center of Excellence.

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

## A   BAYESIAN OPTIMIZATION WITH DEEP RANKING ENSEMBLES

Once the Deep Ensembles are trained, we aggregate the predictions for an input $x$ following the procedure explained in Section 3.2 to obtain $\mu(x), \sigma(x)$ and conditioning to a set of observations $\mathcal{D}_s$. For the sake of simplicity, we omit this conditioning in our notation. These outputs can be fed in several types of acquisition functions and decide for the next point $x$ to observe from the set of pending points to evaluate $\mathcal{X}$. Notice that the lower rank, the better the configuration, therefore we formulate the cast the acquisition function as a minimization problem. Specifically, we consider:

- **Average Rank**: $\alpha(x_j) = \mu(x_j)$
- **Lower Confidence Bound**: $\alpha(x_j) = \mu(x_j) - \beta \cdot \sigma(x_j)$
- **Expected Improvement**: $\alpha(x_j) = -\int_r \max\left(0, \mu(x_k) - r\right) \mathcal{N}\left(r; \mu(x_j), \sigma(x_j)\right)$

Where $\beta$ is a factor that trades of exploitation and exploration and $x_k$ is the best-observed configuration, i.e. $k = \arg\min_{i \in \{1,...,|\mathcal{D}_s|\}} y_i$ and $\mu(x_k)$ is the average rank predicted for that configuration and $y_k$ is its validation error. The previous formulation assumes a minimization, thus to choose the next query point you apply: $x = \arg\min_{x_j \in \mathcal{X}} \alpha(x_j)$.

---

**Algorithm 2:** Bayesian Optimization with DRE

---

**Input**  : A prior distribution over datasets $p(\mathcal{D})$, initial observations
$H = \{(x_1, y_1), ..., (x_N, y_N)\}$, pending points $\mathcal{X}$, number of BO iterations $K$,
black-box function to optimize $f$

**Output:** Best observed configuration $x_*$

1 Train ensemble of MLP scorers following Algorithm 1 and prior $p(\mathcal{D})$;
2 **for** $j \leftarrow 1$ **to** $K$ **do**
3     Fine-tune/Train MLP scorers ;
4     Suggest next candidate $x = \arg\min_{x_j \in \mathcal{X}} \alpha(x_j, H)$ ;
5     Observe response $y = f(x)$ ;
6     Update history $H = H \cup \{(x, y)\}$;
7 **end**
8 Return top performing configuration: $\arg\min_{(x_i, y_i) \in H} y_i$

---

## B   EXPERIMENTAL SETUP FOR DEEP RANKING ENSEMBLES

**Meta-Feature Extractor** The DRE model has two configurable components: the meta-feature network and the scorers. The meta-feature extractor is a DeepSet with an architecture similar to the one used by Jomaa et al. (2021a). However, we used 2 fully connected layers with 32 neurons each for both $\phi$ and $\rho$ (Deep Set parameters) instead of 3 fully connected layers. The output size is set to 16 by default.

**Ensemble of Scorers** The ensemble of scorers is a group of 10 MLP (Multilayer Perceptrons) with identical architectures. Each neural network has 4 hidden layers and each hidden layer has 32 neurons. The neural networks are initialized independently and randomly (for DRE-RI) or warm-initialized with the meta-learned weights. The input size of each neural network is 16 (the dimesiionality of the meta-features), plus the HP search space dimensionality. their output size is 1.

**Setup for Motivating Example.** For the creation of the Figure 2, we use as scorer network an MLP with 2 hidden layers and 10 neurons per layer. The meta-feature extractor has 4 layers and 10 neurons, and output dimensions equal to 10. The network is meta-trained for 1000 epochs, with batch size 10, learning rate 0.001, Adam Optimizer, and 10 models in the ensemble. For the meta-learning example, we do not fine-tune the networks, while we fine-tune the networks for the non-meta-learned example for 500 iterations.

Table 1: Average Cost per BO Step (in seconds)

|          | 4796 (3 Dims)    | 5636 (6 Dims)    | 5527 (8 Dims)    | 5965 (10 Dims)   | 5906 (16 Dims)   |
|----------|------------------|------------------|------------------|------------------|------------------|
| **HEBO** | $0.27 \pm 0.18$  | $3.11 \pm 1.68$  | $2.66 \pm 0.95$  | $3.21 \pm 1.78$  | $2.85 \pm 2.43$  |
| **FSBO** | $10.49 \pm 2.92$ | $10.13 \pm 1.51$ | $10.61 \pm 4.47$ | $11.45 \pm 4.35$ | $12.13 \pm 6.41$ |
| **DRE**  | $22.29 \pm 3.81$ | $18.8 \pm 3.57$  | $22.61 \pm 3.85$ | $19.39 \pm 3.81$ | $22.29 \pm 3.79$ |

## C  DISCUSSION ON LIST SIZE AND LIST WEIGHTS

We present an additional ablation on the list size. We report the average rank on the meta-validation split for different list sizes during meta-training on Figure 8a. Notice, that a small list size (10) leads to an underperforming setting. Therefore, it is important to consider relative large list sizes ($100 \leq n$).

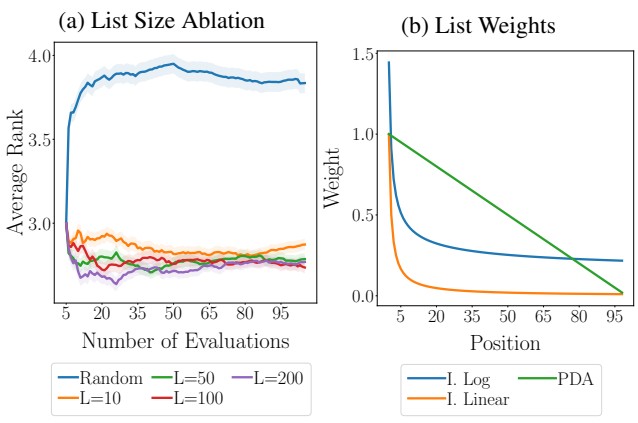

Figure 8: Effect of parameters in list-wise loss

During meta-testing i.e. by performing BO, there is no significant overhead in terms of having a larger list size, because the true rank is derived from the observed validation accuracy of configurations. During both meta-training, as well as the BO step, we fit our surrogate to estimate the rank of previously observed configurations that have been already evaluated. Given $n$ observations, computing the true rank is a simple $\mathcal{O}(n \cdot \log(n))$ sorting operation. Notice that in BO settings $n$ is typically small.

There are several weighting schemes. Two alternatives to the weighting factor we use (inverse log weighting) are inverse linear weighting and position-dependent attention (PDA) (Chen et al., 2017). As you can see in Figure 8b, inverse linear gives very small weight to lower ranks, while the position-dependent gives too much importance. In this plot, PDA weights were scaled to make it comparable to the other schemes. We decided to use the inverse log weighting because it gives neither too low nor too high weight to lower ranks. For the $j$-th position in a list with $k$ elements, these weights can be described as follows:

- **Inverse Log:**  $w(j) = \frac{1}{\log(j+1)}$

- **Inverse Linear:**  $w(j) = \frac{1}{j}$

- **Position-dependent attention:**  $w(j) = \frac{k-j+1}{\sum_{t=1}^{k} t}$

## D  DISCUSSION ON COMPUTATIONAL COST

We provide here a cost comparison between DRE, FSBO and HEBO. In the Table 1, we provide the average cost per BO step ($\pm$ standard deviation) for different search spaces (with different dimensions). DRE effectively incurs a cost higher than FSBO and HEBO, but <30 seconds, which is a very small overhead compared to the cost of actually evaluating hyperparameter configurations (evaluation means the expensive process of training classifiers given the hyperparameter configurations and computing the validation accuracy).

# E   ADDITIONAL PLOTS

We present additional results on the critical difference diagrams for *i)* Transfer methods results (Figure 9a), *ii)* Non-Transfer (Figure 9b, *iii*) Scorer size (Figure 10a, *iv*) Acquisition Function (Figure 10b, *v* Ranking Loss (Figure 11a) and *vi* Meta-features (Figure 11b). These CD plots show the comparison of the performance at different number of trials (e.g. at 25 trials = Rank@25). The vertical lines connecting two methods indicate that their performances are not significantly different.

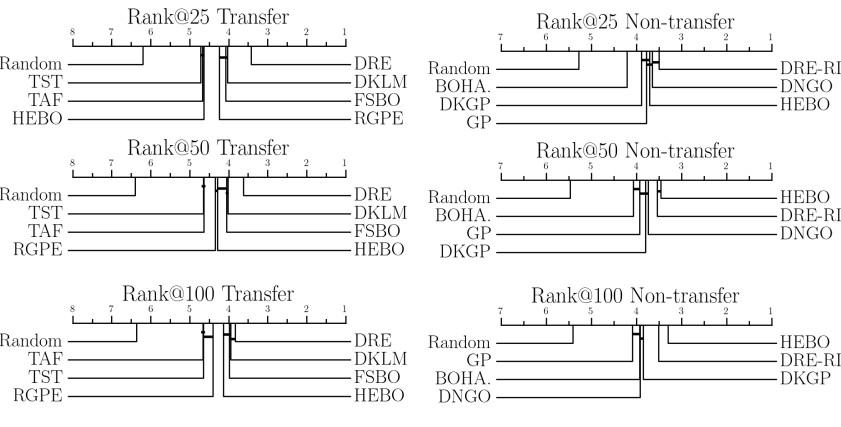

(a) Comparison vs. transfer methods    (b) Comparison vs. non-transfer methods

Figure 9: Critical Difference Diagram for a) Transfer and b) Non-transfer.

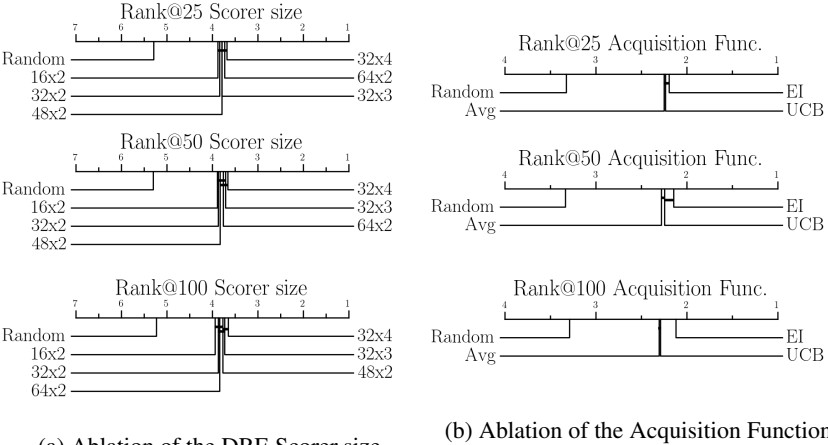

(a) Ablation of the DRE Scorer size

(b) Ablation of the Acquisition Function

Figure 10: Critical Difference Diagram for the results of the ablation of DRE hyperparameters in (a) and the choice of the acquisition function from Hypothesis 5 in (b).

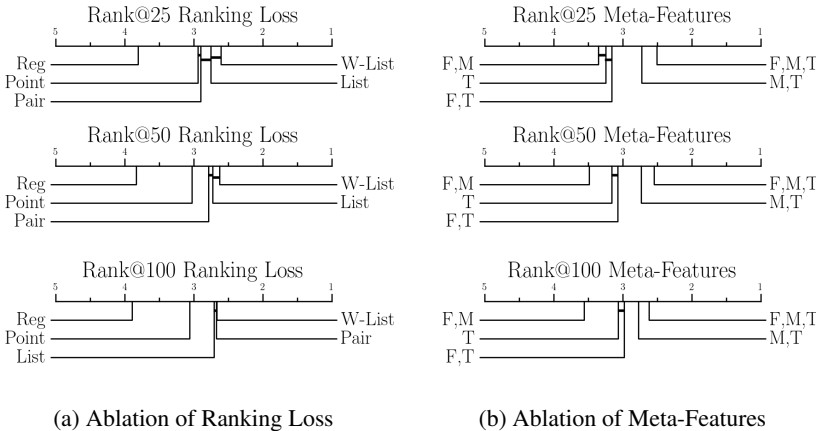

(a) Ablation of Ranking Loss          (b) Ablation of Meta-Features

Figure 11: Critical Difference Diagrams for the results of Hypothesis 3 in a) and Hypothesis 4 in b).

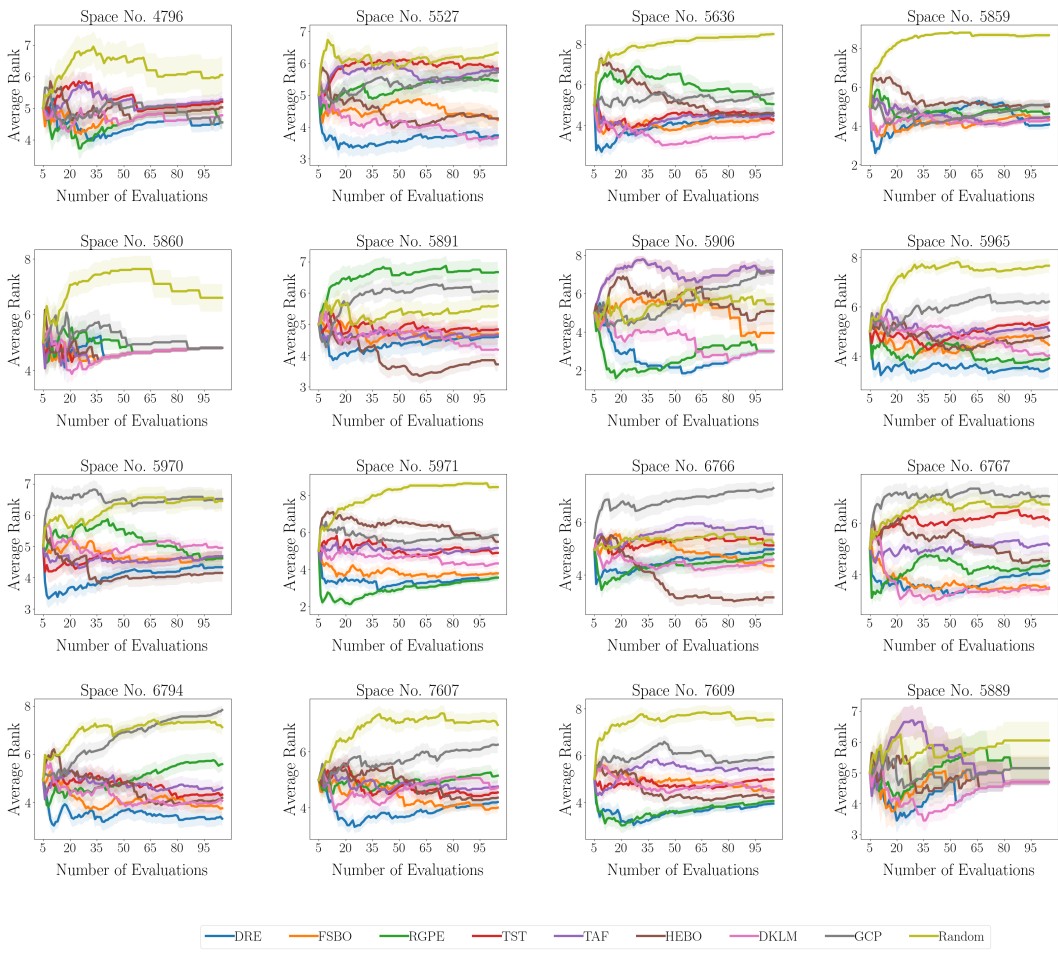

Figure 12: Average Rank per Search Space (Transfer Methods)

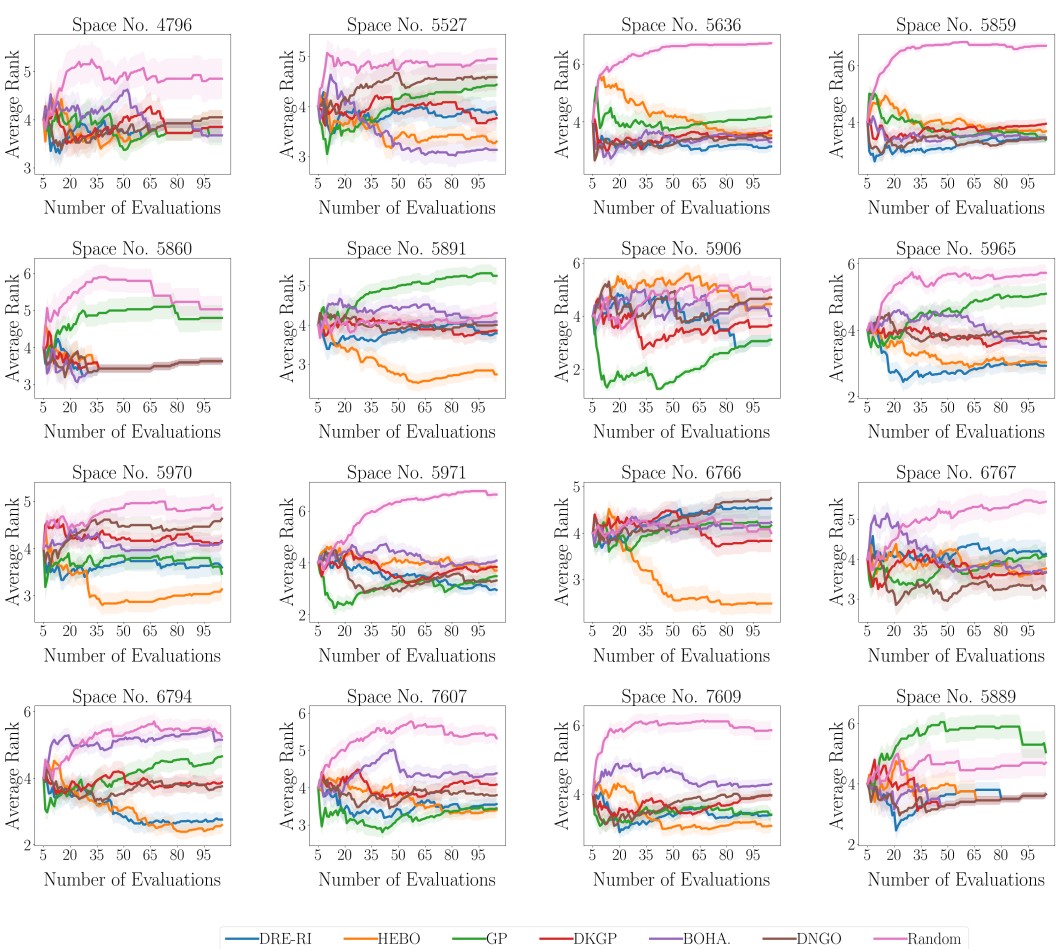

Figure 13: Average Rank per Search Space (Non-Transfer Methods)

