# OpenReview forum: "Deep Ranking Ensembles for Hyperparameter Optimization"
_ICLR.cc/2023/Conference — ICLR 2023 poster_

### Official Review · Reviewer_Eztd · 2022-10-20

**Confidence:** 4
**Correctness:** 4
**Technical Novelty And Significance:** 3
**Empirical Novelty And Significance:** 4
**Recommendation:** 8

**Clarity, Quality, Novelty And Reproducibility:**

The paper is clear, well-written and contains some novel ideas. Experiments are performed on publicly available benchmarks, and code is also provided for the method (although I have not tried to run the code myself).

**Strength And Weaknesses:**

Strengths:
- While not all components here are novel (e.g., the method re-uses deep ensembles for uncertainty quantification and the dataset meta-feature approach from existing works), the core idea of using learning to rank inside BO surrogates is novel to the best of my knowledge, and the combination of this with the existing ideas is innovative and worthy of publication.
- The illustrative example provided really helps the reader to grasp the main concepts, which is much appreciated.
- Extensive experiments are provided comparing with many baselines in both the transfer and non-transfer context, in addition to a variety of different ablation studies, and statistical hypothesis testing is provided in the appendix.

Weaknesses:
- In Section 4.3 it is stated that during the meta-test phase the scoring network is training for 1000 epochs. How sensitive is the algorithm to the number of epochs used? It seems rather a lot, given that in early phase of BO one might only have a handful of configurations that have been evaluated. Are mini-batches used during the online training phase? If so, what was the batch size? If not, how is diversity ensured when training the scoring networks (it is stated in Section 3.2 that diversity in ensured by sampling different mini-batches).
- No mention is provided about the runtime of DRE compared to other baselines. I understand that in the limit when it is very expensive to evaluate a hyperparameter configuration, this may not matter very much. However. it would give a lot of insight into whether this scheme could be used to solve more general classes of black-box optimization problems in which evaluating a configuration may be less expensive than training an ensemble of neural networks for thousands of epochs.
- It was not very clear to me what is the role of the dataset meta-features in the non-transfer setting. Based on last sentence at the end of Section 4.3, it seems that the meta-feature extractor is trained also for DRE-RI. Could the authors comment on how these features might be expected to help in this context?
- I appreciate the authors including the somewhat negative result comparing to HEBO in non-transfer setting (even if it is clearly surpasses by DRE in the transfer setting). Can the authors provide any explanation or intuition why HEBO might be winning in the non-transfer setting when the number of trials gets large.

Minor corrections:
Section 4.3 - I believe the text says "32 networks" when it should say "32 neurons"


**Summary Of The Paper:**

This paper proposes a new approach for hyperparameter optimization (Deep Ranking Ensembles) in both the transfer and non-transfer learning setting.

The are 3 main components to this approach:
1. Use neural networks that learn to rank configurations as a surrogate model in the Bayesian optimization setting.
2. Create diverse ensembles of these networks in order to perform uncertainty quantification (Lakshminarayanan et al., 2017), which is required to use acquisition functions like expected improvement
3. Enhance the surrogate models by using additional learned dataset meta-feature representations (Jomaa et al., 2021).

An illustrative example of the method is provided, followed by an extensive evaluation on the HPO-B benchmark.


**Summary Of The Review:**

This is a well-written paper with some interesting new ideas and impressive experimentation. Given the general applicability of the approach, I think the potential for impact is high.

---

> ### Author Response · Authors · 2022-11-18
> **Reply to Reviewer EzTd**
>
> Dear Reviewer,
>
> thank you for your comments and suggestions. We address your questions as follows:
>
> 1. **On the number of epochs in meta-test and on diversity of the models:** You have an interesting point. Indeed, you would need to train less the network when the number of observations is low, especially if the network is meta-trained. Therefore, it is possible to think about strategies that increase the number of epochs accordingly. We use the whole set of observations during fine-tuning. The diversity is ensured by having different weight initializations among the networks (the deep ensemble paradigm), as proposed by [1].
>
> 2. **On the computational cost**: Please see our general reply or the Appendix D in the updated manuscript.
>
> 3. **On the meta-features when non-transfer:** This is an important point. The meta-features are indeed not very necessary in the non-transfer case (see line “T” and “F,T” on Figure 6b). We included this to have a comparison of the effect of meta-training or not meta-training the very same network (thus including the meta-feature extractor, i.e. line “F,T” vs “F,M,T”).
>
> 4. **On the comparison against HEBO:** As an intuition: The uncertainty estimation of HEBO’s Gaussian Process is hypothetically better at exploring the search space in the later phases of the BO search, compared to the inferior posterior uncertainty of our neural network ensemble. In addition, the multiobjective acquisition function in HEBO is providing even better exploratory capabilities. A still open question in the HPO community is how to ensure that pure neural network based solutions offer qualitative posterior uncertainties to match Gaussian Processes. On the other hand, neural networks (as our method) are superior to GPs in terms of the fit to complex loss surfaces. We hypothesize that the (i) better fitness and (ii) the worse uncertainty estimate lead to our method (i) detecting regions close to the local optima faster, but (ii) exploring less at the later stages. In contrast, exploration is less crucial in the meta-learned version, because we have a strong prior of the loss regions where minima occur (please consult our motivation).
>
> **References**:
>
> [1] Lakshminarayanan, Balaji, Alexander Pritzel, and Charles Blundell. "Simple and scalable predictive uncertainty estimation using deep ensembles." Advances in neural information processing systems 30 (2017).

---

### Official Review · Reviewer_MGU4 · 2022-10-24

**Confidence:** 3
**Correctness:** 4
**Technical Novelty And Significance:** 3
**Empirical Novelty And Significance:** 3
**Recommendation:** 8

**Clarity, Quality, Novelty And Reproducibility:**

The paper is written clearly and with good quality.

In terms of novelty, although major components rely on different works (wight of loss function, uncertainty estimation, transfer learning), but it's a pretty novel idea to combine them all together and use it an a new method for HPO.

**Strength And Weaknesses:**

Strength:
- The idea presented is well motivated with clear mention to related work and background.
- The method is proposed clearly.
- The demonstration of the toy example is very intuitive.
- Experiments are thorough and discussions on the 5 hypotheses are clear.

Weaknesses:
Not something significant but I have several questions for discussion:
- The weight in loss function (1) seems to be from an older paper, and is not tuned (in experiment seems only compare with the non-weighted version)? Do we think there could be better weights available, especially for different tasks?
- The rank of the configurations is a less well behaved surface to model compared to the original loss. Even if the underlying loss function is smooth, the rank could still be of large variability. Intuitively, how the method is able to perform better in this case? Consider a example where there are two local extremes, then modeling the original surface should be fine, but modeling the ranking can be harder due to the similarities of ranks near two local extremes?
- The authors should discuss the efficiency of the algorithm compared to its competitors.

**Summary Of The Paper:**

This paper presents a new idea on hyperparameter optimization (HPO) by formulating it as a weighted learning to rank problem. By using a novel loss function, coupled with existing work on neural ensembles and transfer learning, the proposed method is shown to achieve state-of-the-art results in HPO.

**Summary Of The Review:**

Overall I believe this paper propose a new idea in the important area of HPO, which is proved through extensive benchmark datasets to yield state-of-the-art results. I think it's a good paper for ICLR.

---

> ### Author Response · Authors · 2022-11-18
> **Reply to Reviewer MGU4**
>
> Dear Reviewer,
>
> thank you for your comments and suggestions. We address your questions as follows:
>
> 1. **On the weighting scheme for the list-wise loss:** There are indeed several weighting schemes. Two alternatives to the weighting factor we use (inverse log weighting) are inverse linear weighting and position-dependent attention (PDA) [1]. As you can see in Figure 8a in our updated manuscript, the inverse linear scheme gives very small weight to lower ranks, while PDA weights give too much importance. We decided to use inverse log weighting because it gives neither too low nor too high weight to lower ranks.
>
> 2. **On the rank of local extremes:** Our toy example shows the case where we have two global optima, and still our method outputs an acquisition function that resembles the fact that these points are important to explore. These would be the local extremes, you might refer to.
>
> 3. **On the computational cost:** Please see our general reply above or Appendix D in the updated manuscript.
>
>
> References:
>
> [1] Chen, Huadong, et al. "Top-Rank Enhanced Listwise Optimization for Statistical Machine Translation." arXiv preprint arXiv:1707.05438 (2017).

---

> > ### Comment · Reviewer_MGU4 · 2022-11-21
> > **Thanks for the response.**
> >
> > The authors have addressed my concerns clearly.

---

### Official Review · Reviewer_7HGf · 2022-10-30

**Confidence:** 4
**Clarity, Quality, Novelty And Reproducibility:** 1. Clarity
**Correctness:** 3
**Technical Novelty And Significance:** 3
**Empirical Novelty And Significance:** 3
**Recommendation:** 6

**Strength And Weaknesses:**

**Strengths**:

1. To the best of my knowledge, the ranking-based HPO method is quite novel.
2. A comprehensive evaluation is done to show the effectiveness of the proposed method.

**Concerns and questions**:

1. My main concern about this work is the impact of the ranking list size. I do not find the actual definition of $N^{(s)}$ in Alg.1, but it seems to be the size of the ranking list in the rank scorer. In my opinion, this ranking list size is a crucial factor in building the ranking surrogate. Presumably, a small-size list is not that informative (one extreme case is the size being 1), and a larger size shall be preferred. However, larger size also means that the HPO algorithm needs to evaluate a large batch of configurations at each iteration (according to my understanding, computing the true rank requires one to evaluate the actual performance of the configs in the list). Thus I believe it is important to investigate the impact of this factor in this method. Currently, there is no discussion and no evaluation of this factor.

2. A minor comment on the main hypothesis, "the optimal strategy for training surrogates is to preserve the ranks of the performances of HPO as an L2R problem.": Although the idea of using L2R to learn the ranking of HPs to help with HPO is a reasonably good idea and can be effective, I do not see it necessarily being an "optimal" strategy. Since in HPO, the ultimate objective is to find one config which has the best performance, does the full ranking necessarily matter? This work verifies some hypotheses but does not discuss thoroughly on this main hypothesis.

3. A question on the computation cost: I wonder does training the scorer, and the neural network surrogates bring significant computation overhead to the HPO process. The empirical evaluation in this work only evaluates the performance regarding the number of trials and does not evaluate the end-to-end computation cost. I hope the authors can provide more information on the computation overhead of the rank-based surrogate and also the anytime performance over computation resources used instead of the number of trials.

4. On the meaning of BO iterations, trials, and the actual number of evaluated configurations: I have confusion on (1) Does # of BO iterations = # of trials? (2) Does # of trials in the Figure 5 = # of actual hyperparameter configurations evaluated?

**Summary Of The Paper:**

This work proposes an HPO method that learns to select the best HP from the rankings of HPs. The main insight comes from the authors that the optimal strategy for training surrogates is to preserve the ranks of the performances of HPO as an L2R problem. More specifically, the method meta-learns neural network surrogates optimized for ranking the configurations' performance while modeling their uncertain via ensembling. The ranking surrogate will be used in the BO framework to help with HP suggestion.

**Summary Of The Review:**

This work proposes an HPO method that learns to select the best HP from the rankings of HPs.  To the best of my knowledge, the ranking-based HPO method is quite novel. A comprehensive evaluation is done to show the effectiveness of the proposed method. However, I have concerns about how sensitive the method is on the ranking list size. I also have confusions about how the evaluation is done (see the Strength And Weaknesses).

---

> ### Author Response · Authors · 2022-11-18
> **Reply to Reviewer 7HGf**
>
> Dear Reviewer,
>
> thank you for your comments and suggestions. We address your questions as follows:
>
> 1. **On the list size**:
> - You have an interesting remark. During meta-training we fixed the list size to 100. However, given your question, we performed an ablation of the list size and found out that your intuition is correct. A small list size of 10 underperforms compared to larger list sizes. The results indicate that our actual setting of 100 is a very competitive setting. See Appendix C in our updated manuscript.
> - Regarding the other point you mention, there is no significant overhead in terms of having a larger list size, because the true rank is derived from the observed validation accuracy of configurations. During both meta-training, as well as the BO step, we fit our surrogate to estimate the rank of previously observed configurations that have been already evaluated. Given $n$ observations, computing the true rank is a simple $\mathcal{O}(n\cdot \mathrm{log}(n))$ sorting operation. Notice that in BO settings $n$ is typically small. We stress that the choice of the list size is a concern only for the meta-learning step. In contrast, when conducting HPO on a new dataset, we have a limited number of configurations (from 5 to 100) and our list size is always the cardinality of the set of all observed configurations.
>
> 2. **On the optimality of ranking losses:** In HPO we aim at optimizing a special criterion, finding the top-1 hyperparameter configuration with the highest validation accuracy. In that sense, an “optimal” search procedure is one that minimizes a differentiable proxy function to the non-differentiable top-1. A list-wise loss with weights favoring the top-ranked configuration is therefore a more direct proxy for top-1, yielding our characterization of “optimal” loss. Optimality here means that minimizing the proxy yields a direct minimization of the true criterion. Even though we did not present a theoretic proof of the convergence bounds (in our understanding it is very challenging to theoretically prove the convergence of a complex BO procedure given a specific surrogate loss), we empirically support this position with our ablation of the loss function (Figure 6.a).
>
> 3. **On the computational cost**: See please our general reply above and Appendix D.
>
> 4. **On the meaning of BO iterations**: Thanks for pointing that out. Yes, to both questions. The HPO community uses the term hyperparameter configuration trial more, while the general black-box optimization community often uses the term BO iteration/step. However, it refers to the number of evaluations. We modify accordingly the plots in our updated manuscript.

---

### Author Response · Authors · 2022-11-18
**General Reply**

We thank the reviewers for their feedback, suggestions, and remarks. After your comments, we updated our manuscript with the following changes:

- Table 1 with the computational cost measurements of DRE compared with FSBO and HEBO. See Appendix D and the extended comment below this reply.
- A discussion on the list size and the weighting scheme for the listwise ranking loss. See Appendix C.
- An update to all the plots to reflect the number of evaluations for the sake of clarity. Now the plots start from 5 evaluations, i.e. 5 initial observations, the same for all the methods.

**On Computational Cost**

We provide here a cost comparison between DRE, FSBO, and HEBO. In the table, we provide the average cost per BO step (+/- standard deviation) in *seconds* for different search spaces (with different dimensions). DRE effectively incurs a cost higher than FSBO and HEBO, but <30 seconds, which is a very small overhead compared to the cost of actually evaluating hyperparameter configurations (evaluation means the expensive process of training classifiers given the hyperparameter configs and computing the validation accuracy).

|      |  4796 (3 Dims) |  5636 (6 Dims) |  5527 (8 Dims) | 5965 (10 Dims) | 5906 (16 Dims) |
|------|:--------------:|:--------------:|:--------------:|:--------------:|:--------------:|
| HEBO |  0.27 +/- 0.18  |  3.11 +/- 1.68  |  2.66 +/- 0.95  |  3.21 +/- 1.78 |  2.85 +/- 2.43 |
| FSBO |  10.49 +/ 2.92 | 10.13 +/- 1.51 | 10.61 +/- 4.47 | 11.45 +/- 4.35 | 12.13 +/- 6.41 |
|  DRE | 22.29 +/- 3.81 |  18.8 +/- 3.57 | 22.61 +/- 3.85 | 19.39 +/- 3.81 | 22.29 +/- 3.79 |

---

### Decision · Program_Chairs · 2023-01-20

**Decision:**

Accept: poster

**Justification For Why Not Higher Score:**

Just going for the reviewer opinions. This seems like a good idea with some potential, but the real impact on HPO has not been fully demonstrated. The paper would be stronger by clearly demonstrating not just that ranking based surrogates could be a good idea,
but that they overcome a blocker for conventional surrogate models (for example, high-dimensional and/or conditional search spaces).


**Justification For Why Not Lower Score:**

Going for reviewer opinions.


**Metareview: Summary, Strengths And Weaknesses:**

In model-based HPO, a surrogate model represents the knowledge about optimal configurations which can be obtained from a small number of evaluations. In BO, this is usually done by a probabilistic model of the criterion itself (e.g., validation error). In this work, the authors argue that a more robust model is obtained by just preserving the ranks of the evaluated configurations. Their surrogate is a deep ensemble of neural ranking models.

According to reviewers, this idea is novel and valuable, and the experiments are well-done, moreover the paper is clearly written. It constitutes a useful and interesting addition to HPO research. On the downside, the method is rather slow proposing new configurations, and the stability of retraining neural models after receiving few new observations (which is effectively the continual learning problem) remains a general concern for neural network surrogates applied to BO, so that the proposed method is probably most valuable in the transfer regime, where data from related HPO tasks is available for offline training.


**Note From Pc:**

if the above contains the word "oral" or "spotlight" please see: "oral" presentation means -> notable-top-5% and "spotlight" means -> notable-top-25%. As stated in our emails, we are disassociating presentation type from AC recommendations